# Symmetry regimes for circular photocurrents in monolayer MoSe$_2$

Jorge Quereda[1], Talieh S. Ghiasi[1], Jhih-Shih You[2], Jeroen van den Brink[2],
Bart J. van Wees[1] & Caspar H. van der Wal[1]

In monolayer transition metal dichalcogenides helicity-dependent charge and spin photo-currents can emerge, even without applying any electrical bias, due to circular photogalvanic and photon drag effects. Exploiting such circular photocurrents (CPCs) in devices, however, requires better understanding of their behavior and physical origin. Here, we present symmetry, spectral, and electrical characteristics of CPC from excitonic interband transitions in a MoSe$_2$ monolayer. The dependence on bias and gate voltages reveals two different CPC contributions, dominant at different voltages and with different dependence on illumination wavelength and incidence angles. We theoretically analyze symmetry requirements for effects that can yield CPC and compare these with the observed angular dependence and symmetries that occur for our device geometry. This reveals that the observed CPC effects require a reduced device symmetry, and that effects due to Berry curvature of the electronic states do not give a significant contribution.

[1] Zernike Institute for Advanced Materials, University of Groningen, 9747 AG Groningen, The Netherlands. [2] Institute for Theoretical Solid State Physics, IFW Dresden, Helmholtzstrasse 20, 01069 Dresden, Germany. Correspondence and requests for materials should be addressed to J.Q. (email: j.quereda.bernabeu@rug.nl)

Among two-dimensional (2D) materials, monolayer transition metal dichalcogenides (1L-TMDCs) offer a versatile platform for the development of spintronic and valleytronic devices, where the spin and valley degrees of freedom are used as information carriers[1–5]. The particular band structure of 1L-TMDCs, where two nonequivalent valleys appear at the $K$ and $K'$ points of the 2D Brillouin zone, gives rise to valley-dependent optical selection rules. Specifically, when a 1L-TMDC is illuminated with circularly polarized light with a photon energy close to its bandgap, optical transitions can only take place in one of the two valleys, either $K$ or $K'$, depending on the helicity of the circular polarization, leading to a light-induced valley population imbalance[5]. Additionally, monolayer TMDCs present a large spin-orbit splitting, which sign changes between the $K$ and $K'$ valleys, causing a coupling between the spin and valley degrees of freedom[5]. As a consequence, different optical processes such as the valley Hall effect[6,7] can be used to generate spin and valley polarized photoresponse in TMDCs. For this effect, under circularly polarized illumination, charge carriers in different valleys flow to opposite transverse edges when an in-plane electric field is applied, producing a light helicity-dependent Hall voltage.

The recent observation of helicity-sensitive circular photogalvanic effect (CPGE)[8,9], both for multilayer and monolayer TMDCs[10,11], opens another route for producing spin–valley transport through a 1L-TMDC phototransistor. Differently from the valley Hall effect, which relies on applying an in-plane voltage gradient to the TMDC in order to obtain spin–valley current, the CPGE allows to generate a directed spin–valley current even without applying any voltage, bringing new opportunities for the implementation of spintronic and valleytronic devices where the direction and intensity of spin and valley currents can be controlled using light only. However, a comprehensive study of this effect and its microscopic origin in 1L-TMDCs is still missing.

In this work, we investigate the spectral and electrical behavior of the helicity-dependent circular photocurrent (CPC) in a 1L-TMDC, providing a comprehensive experimental characterization of this effect. In an earlier work[11], it was suggested that exciton transitions could play a role in the generation of CPC. Here, by evaluating the spectral response of the CPC in a h-BN-encapsulated 1L-MoSe$_2$ phototransistor, we show that the CPC amplitude is maximized when the illumination wavelength matches the A exciton resonance, clearly confirming the excitonic character of CPC in 1L-TMDCs. In another recent experiment on multilayer WSe$_2$[10], it was shown that the strength of the CPC response could be changed by a gate voltage, but the effect of the drain-source voltage $V_{ds}$ was not studied. Our study here includes the dependence of the CPC on $V_{ds}$, revealing two different regimes for small (below 0.4 V) and large voltages, with the CPC changing sign between one regime and the other. At certain fixed $V_{ds}$ this CPC sign change can also be induced via the gate voltage $V_{gate}$. Further, by testing the dependence of CPC on the light incident angle we find that it presents very different symmetry for the two regimes: for small $V_{ds}$, the CPC is preserved when the incidence angle is switched from $\phi$ to $-\phi$, whereas for large $V_{ds}$, inverting the illumination angle $\phi$ causes a change of sign for the CPC, pointing to a separate physical origin. In ref.[11] it was proposed that Berry curvature (BC) at the band edges of 1L-TMDCs can give a contribution to CPC (BC-induced CPGE, BC-CPGE). However, the expected dependence of this effect on the light incidence angle is not compatible with the angular dependences observed here for any of the two CPC regimes. Thus, we conclude that BC-CPGE can be ruled out as a dominant mechanism involved. Further, we show that CPC can also emerge in this system due to the circular photon drag effect (CPDE), mostly overlooked in prior literature for 1L-TMDCs. Finally, we show how by applying a gate voltage to modify the Fermi energy of the 1L-MoSe$_2$ channel, one can tune the relative strength of the two contributions at a fixed drain-source voltage, achieving control over the intensity and direction of the helicity-dependent photoresponse.

## Results

**CPC in 1L-TMDC for interband transitions.** When spatial inversion symmetry is broken in the 2D plane in a system with time-reversal symmetry, illumination with circular light can generate a DC photocurrent $\vec{J}$ that behaves as a second-order response to the electric field. $\vec{J}$ can be written as a series expansion in the light wave vector $\vec{q}$ as $J_l = \chi_{ljk} E_j E_k^\star + T_{ljk} q_\mu E_j E_k^\star + (\ldots)$. Here $\chi_{ljk}$ and $T_{ljk}$ are the photogalvanic and photon drag susceptibility tensors and $l$, $j$, $k$, and $\mu$ label Cartesian coordinates $x$, $y$, and $z$. As we present in Supplementary Note 6, the device symmetries strongly constrain the tensor components, and they can still vanish for high-symmetry configurations, even for broken inversion symmetry. We consider three different symmetry scenarios: $D_{3h}$ (pristine 1L-MoSe$_2$), $C_{3v}$ (1L-MoSe$_2$ with broken out-of-plane mirror symmetry), and single-mirror symmetry (1L-TMDCs in the presence of strain or device inhomogeneities). Comparing the dependence of CPC on illumination angles with the symmetry-allowed CPGE and CPDE contributions, we find that for the low-bias regime our observations are only compatible with a device symmetry of, at most, a single-mirror plane. For the high-bias regime the CPC effects are also compatible with $C_{3v}$ symmetry.

In previous reports, the CPC in 1L-TMDCs has been associated with a BC-induced CPGE[11]. In 1L-TMDCs, the BC takes opposite signs at the $K$ and $K'$ valleys, giving rise to counterpropagating valley currents[5,6,12]. Thus, when circularly polarized illumination is used to produce a valley population imbalance, a CPC contribution can appear. In Supplementary Note 6 we derive the CPGE photocurrent using the Fermi Golden rule. This shows that resonant interband transitions can produce a BC contribution to the CPGE[13]. However, this contribution should maximize for incidence perpendicular to the 2D plane[14], while our experiments only show nonzero CPC at oblique incidence (see below).

**Device fabrication, electrical characterization, and setup.** The fabricated 1L-MoSe$_2$ field effect transistor is depicted in Fig. 1a and the actual device is shown in Supplementary Note 2. To improve the device quality and stability[15,16], the 1L-MoSe$_2$ channel is encapsulated between a bilayer and a bulk h-BN flake, acting as the top and bottom layers, respectively. The 2L-BN/1L-MoSe$_2$/bulk-BN stack is directly built onto a SiO2/doped Si substrate with an oxide thickness of 300 nm. The electrodes are fabricated on top of the structure by e-beam lithography and e-beam evaporation of Ti (5 nm)/Au (55 nm) (see Methods section). To further avoid the presence of adsorbates and contaminants, the sample is kept in vacuum ($10^{-4}$ mbar) during the whole experiment. All experiments were carried out at room temperature.

Electrical characterization of the sample (Supplementary Note 3) clearly shows the $n$-type character of the 1L-MoSe$_2$ channel, with a threshold gate voltage of about 20 V and an electron mobility of 17 cm$^2$ V$^{-1}$ s$^{-1}$. In this sample geometry, the bilayer h-BN layer plays the role of a tunnel barrier, preventing Fermi level pinning at the metal–semiconductor interfaces[17].

**Helicity-resolved photovoltage measurements: description and phenomenological formula.** Figure 1a depicts the experimental setup for measuring the helicity-dependent photogalvanic response of the MoSe$_2$ phototransistor. We illuminate the sample

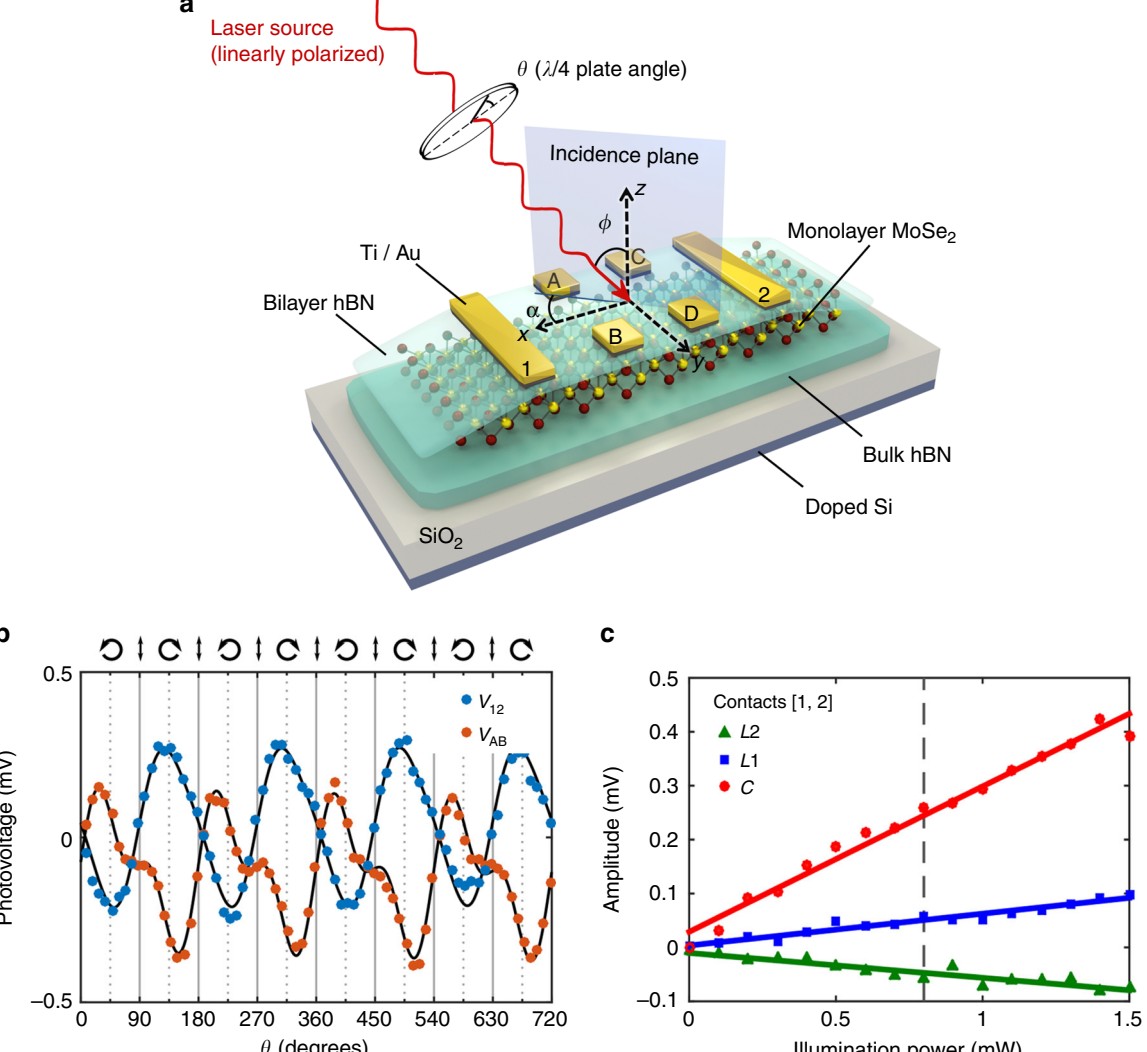

**Fig. 1** Experiment geometry and helicity-dependent response. **a** Schematic experimental setup. The helicity of the laser excitation is controlled by rotating the quarter-waveplate angle, $\theta$. **b** Helicity-dependent photovoltage of the contacts [1, 2] (blue) and [A, B] (orange) as a function of the quarter-waveplate angle $\theta$ for $\lambda = 785$ nm, $\phi = 20°$, $V_{ds} = 0$, $V_{gate} = 0$, and $\alpha = 45°$. The black lines are fits to the phenomenological equation (1). **c** Power dependence of $C$, $L1$, and $L2$ (extracted from fits to Eq. (1)). The solid lines are linear fits to the experimental data. The vertical dashed line indicates the power used during the experiments, 0.8 mW

at an oblique angle $\phi$ with respect to the normal vector of the crystal surface and simultaneously measure the photoinduced currents, either directly (Supplementary Note 5) or as the associated voltages (main text). We used two perpendicular sets of electrodes, giving voltage signals $V_{12}$ and $V_{AB}$. For illumination, we used a laser with tunable photon energy. For achieving a uniform illumination power density and well-defined light incidence angles, we used a collimated beam of 0.5 cm diameter, much larger than the studied device. The polarization of the illumination beam was tuned by rotating a $\lambda/4$ waveplate over an angle $\theta$: during rotation over 360° the original linear polarization gets modulated twice between left and right circular polarization (see top labels Fig. 1b).

Figure 1b shows $V_{12}$ and $V_{AB}$ as a function of $\theta$ for illumination fixed at 785 nm (1.58 eV, on-resonance with the $A^0$ exciton transition of monolayer MoSe$_2$[2,18,19]), incidence angle $\phi = 20°$, and azimuthal angle $\alpha = -45°$ (defined as the angle between the $x$-axis and the incidence plane, see Fig. 1a). The gate voltage was fixed to $V_{gate} = 0$ V. Both voltages clearly show a polarization dependence, with $2\theta$-periodic and $4\theta$-periodic components. The fingerprint of a CPC contribution is its helicity

dependence, appearing as a signal $V_{CPC} \propto \sin(2\theta)$. A $4\theta$-periodic modulation, $V_{LPC}$, can also appear due to linear photogalvanic and linear photon drag effects[8]. The total photovoltage $V_{PC}$ can be described phenomenologically as[8,11,20,21]

$$V_{PC} = V_0 + C\sin(2\theta) + L_1\sin(4\theta) + L_2\cos(4\theta), \quad (1)$$

where $C$ accounts for the CPC and $L_1$ and $L_2$ account for the linear photogalvanic and photon drag effects. The total linear polarization-dependent contribution can be accounted as $L = \left(L_1^2 + L_2^2\right)^{1/2}$. An additional polarization-independent term, $V_0$ (typically smaller or, at most, comparable to $C$), can also appear due to inhomogeneities or thermal drifts between the two electrodes. We obtain values for $C$, $L_1$, and $L_2$ by fitting Eq. (1) to data as in Fig. 1b. Figure 1c shows the power dependence of $C$, $L_1$, and $L_2$. The three amplitudes increase linearly with the illumination power $P$, confirming that they are due to a second-order response to the light electric field $E$ (and thus, proportional to $E^2 \propto P$), in agreement with earlier literature for 1L-MoS$_2$[11].

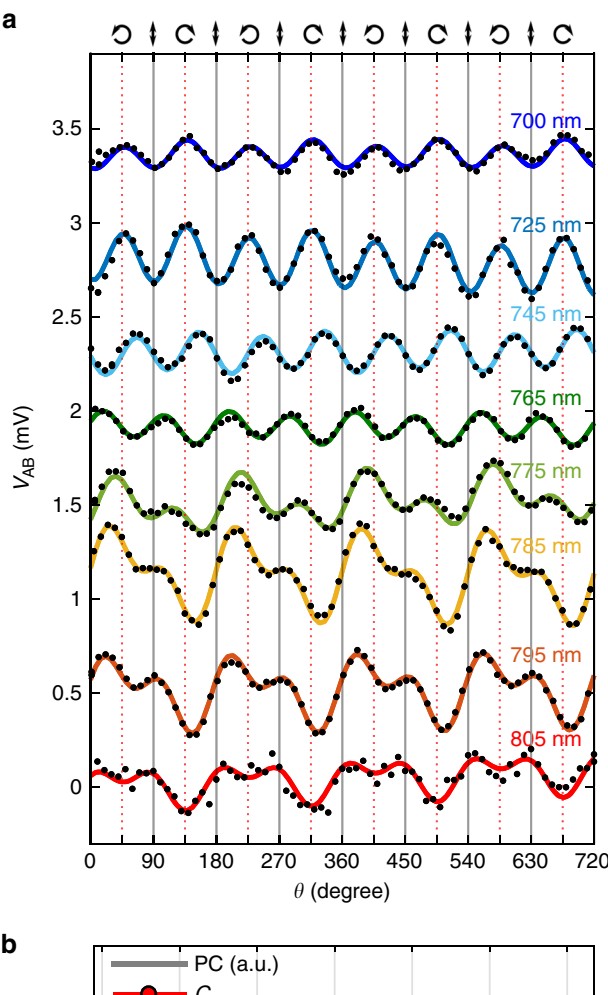

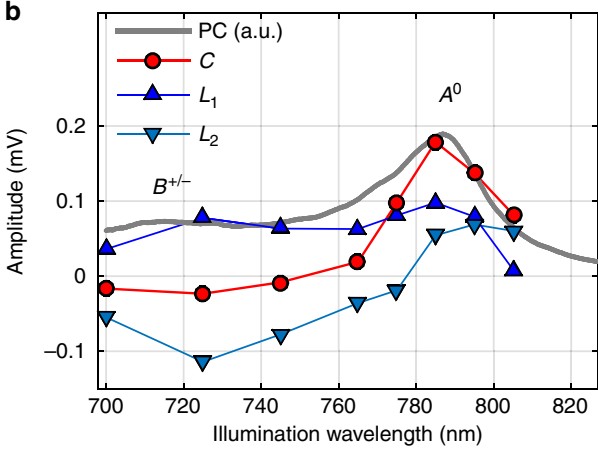

**Fig. 2** Spectral evolution of the circular photocurrent. **a** $V_{AB}$ as a function of the waveplate angle, $\theta$ (for $\phi = 20°$, $V_{ds,12} = 0$, $V_{gate} = 0$ and $\alpha = 45°$) under different illumination wavelengths, from 700 nm to 825 nm. For clarity, the traces have been vertically shifted in steps of 0.5 mV. The solid lines are fits to Equation 1. **b** Photocurrent spectrum of the 1L-MoSe$_2$ crystal (grey, solid line) and spectral dependence of the fitting parameters $C$, $L_1$ and $L_2$ (red, dark blue and pale blue lines, see legend)

Finally, we remark that the CPC signal $C$ behaves as reported below for multiple electrode configurations. We can thus rule out that our CPC signals emerge due to properties of specific contacts, or effects from confinement of light between the micron-scale metallic electrodes. We elaborate on this in Supplementary Note 5.

**Spectral response of the CPC**. Next, we investigate the spectral response of the observed helicity-dependent photovoltage. We first characterize the spectral features of the monolayer MoSe$_2$ phototransistor by photocurrent spectroscopy[2] (see ref.[19] for detailed discussion about our measurement technique). We illuminate the sample using a linearly polarized continuous-wave tunable infrared laser and register the photovoltage as a function of the illumination wavelength at a constant drain-source bias, $V_{ds} = 1$ V. The resulting photocurrent spectrum (gray line in Fig. 2b) shows a prominent peak at 1.58 eV (785 nm), corresponding to the $A^0$ exciton resonance of MoSe$_2$. Trion absorption ($A^{+/-}$, resonant for 795 nm) possibly gives here a small parallel contribution. A second, less prominent, peak can also be observed at 1.74 eV (713 nm), which results from the $B^{+/-}$ trion transition[2,19]. For further comment on exciton and trion transitions in 1L-MoSe$_2$ we address the reader to Supplementary Note 8.

Figure 2a shows the helicity-dependent photovoltage of the 1L-MoSe$_2$ device and fits to Eq. (1) for different illumination wavelengths. Figure 2b shows the wavelength dependence of $C$, $L_1$, and $L_2$. The CPC contribution $C$ is maximal when the illumination is on-resonance with the $A$ exciton or trion transitions ($\lambda = 785–795$ nm) and progressively decreases when the illumination is shifted away from the resonance. For the linear photovoltage $L$, a nonzero amplitude appears, also for out-of-resonance illumination. Further, we observed that the spectral dependence of $L$ markedly changes between different sets of electrodes, even in the same 1L-MoSe$_2$ flake. The origin of a nonzero $L$ is usually associated with scattering of the charge carriers with anisotropic local defects[20]. In our system, an additional contribution to $L$ could arise from plasmonic effects, since the distance between the contacts is comparable with the illumination wavelength range. These effects are expected to be very sensitive to the contact geometry (further discussed in Supplementary Note 5).

The observed spectral behavior of $C$ shows that interband excitons play a central role in the CPC photoresponse. However, since excitons are charge-neutral quasiparticles, they must dissociate to produce a nonzero photocurrent. The required dissociation can be assisted by the large in-plane electric fields present in the depletion regions near a metal–semiconductor junction, especially when a bias voltage is applied[7]. Alternatively, a photocurrent can appear in the absence of in-plane electric fields if trions are present in the MoSe$_2$, since they have a nonzero net charge, and can contribute to the photocurrent even without dissociating. Since in our system the whole device is illuminated, both dissociated excitons and non-dissociated trions are expected to play a role in the CPC.

**Effect of a nonzero drain-source voltage**. To investigate the influence of an in-plane electric field on the CPC we apply a drain-source voltage $V_{ds}$ between the electrodes $A$ and $B$ and measure the transverse voltage between the electrodes 1 and 2, while keeping $\lambda = 785$ nm, $V_{gate} = 0$, $\phi = 20°$, and $\alpha = 45°$. For improving the signal-to-noise ratio, we now use a chopper to modulate the laser intensity at 331 Hz and lock-in detection of the AC photovoltage $V_{12}^{AC}$. Figure 3a, b show the helicity dependence of $V_{12}^{AC}$ at different $V_{ds}$ and the associated dependence of $C$ and $L$ on $V_{ds}$. Unlike the case of the valley Hall effect (where the anomalous Hall voltage changes linearly with the applied drain-source voltage), the CPC response observed here preserves its sign when the direction of the drain-source voltage is inverted. For small applied voltages, up to $|V_{ds}| \sim 0.4$ V $\equiv V_T$ (transition voltage), $C$ remains constant. When increasing $|V_{ds}|$ above $V_T$ the

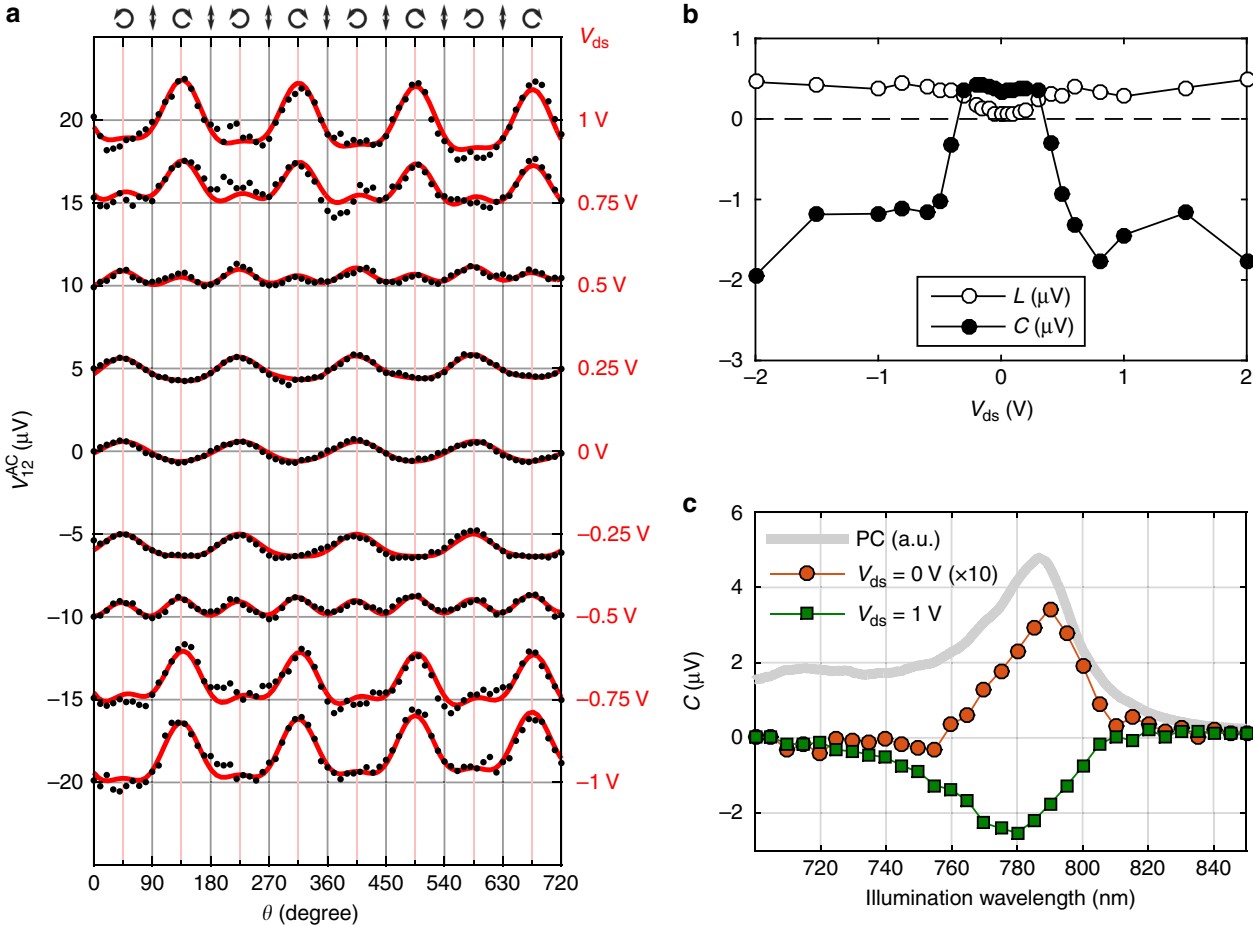

**Fig. 3** Helicity-dependent photovoltage, $V_{12}^{AC}$ for different drain-source voltages $V_{ds}$ and with $\lambda = 785$ nm, $\phi = 20°$, $V_{gate} = 0$, and $\alpha = 45°$. **a** $V_{12}^{AC}$ as a function of $\theta$ for different drain-source voltages. For clarity, the measurements have been vertically shifted in steps of 5 μV and the polaritazion-independent offset, $V_O$, has been substracted (see Eq. (1)). **b** $C$ and $L$ parameters as a function of the drain-source voltage. **c** CPC amplitude, $C$, as a function of the wavelength for $V_{ds} = 0$ V (orange circles) and $V_{ds} = 1$ V (green squares). For an easier visualization, the data for $V_{ds} = 0$ V have been multiplied by 10

photogalvanic signal undergoes an abrupt change of sign and becomes ~5–10 times larger.

Figure 3c shows the CPC amplitude $C$ as a function of the wavelength at $V_{ds} = 0$ V and $V_{ds} = 1$ V. Interestingly, the wavelength at which $C$ is maximized (in absolute value) for $V_{ds} = 0$ V occurs at a wavelength 5–10 nm longer (in energy 10–20 meV lower) than that obtained for $V_{ds} = 1$ V. A similar spectral shift is also observed for different gate voltages (see Supplementary Note 7). The observed shift is consistent with recent photoluminescence experiments that showed a gate-induced exciton-to-trion transition in a monolayer TMDC[22]. This suggests that at low drain-source voltages the dominant charge carriers involved in the CPC are $A^{+/-}$ trions (which have a nonzero charge and therefore do not need to dissociate to participate in the photovoltage), while at large drain-source voltages the transport is dominated by dissociated $A^0$ excitons. Note that, given the $n$-type behavior of the 1L-MoSe$_2$ channel, negatively charged trions $A^-$ are expected to have a much larger contribution to the CPC than their positive counterparts $A^+$. We also remark that, as shown in recent experiments, dark-exciton transitions in TMDCs can become optically active under oblique illumination[23]. Further, earlier work by our team showed that for a non-encapsulated 1L-MoSe$_2$ device dark-exciton contributions can even appear under illumination perpendicular to the crystal

plane[24], giving an additional peak in the PC spectrum at an energy ~30 meV above the $A^0$ resonance, that is, at ~775 nm. This feature is not observed with the h-BN-encapsulated device studied here, suggesting that the asymmetric interaction with SiO$_2$ could be relevant to opening the dark-exciton optical transition. h-BN substrates are known to yield a strengthened $A^0$ exciton absorption[25], which could mask a weak dark-exciton contribution in the present work.

**Effect of the illumination angle in the CPC.** In order to identify the symmetry properties of the two different CPC regimes (for $V_{ds}$ above and below $V_T$), we test their behavior under different illumination angles. Figure 4 shows the measured helicity-dependent photovoltage $V_{12}^{AC}$ for different illumination incidence angles, $\phi$, in the low $V_{ds}$ (Fig. 4a) and high $V_{ds}$ (Fig. 4b) regimes. Remarkably, these two regimes show a very different behavior: for $V_{ds} = 0$ V, the CPC shows the same sign and a similar amplitude at $\phi = 20°$ and $\phi = -20°$, while, for $V_{ds} = 1$ V, inverting the angle of incidence causes the CPC to reverse its sign, pointing to two separate physical mechanisms. Importantly, for both situations $C$ vanishes for incidence normal to the 2D plane, $\phi = 0°$, which rules out that BC-induced CPC gives significant contributions to our signals (Supplementary Note 6).

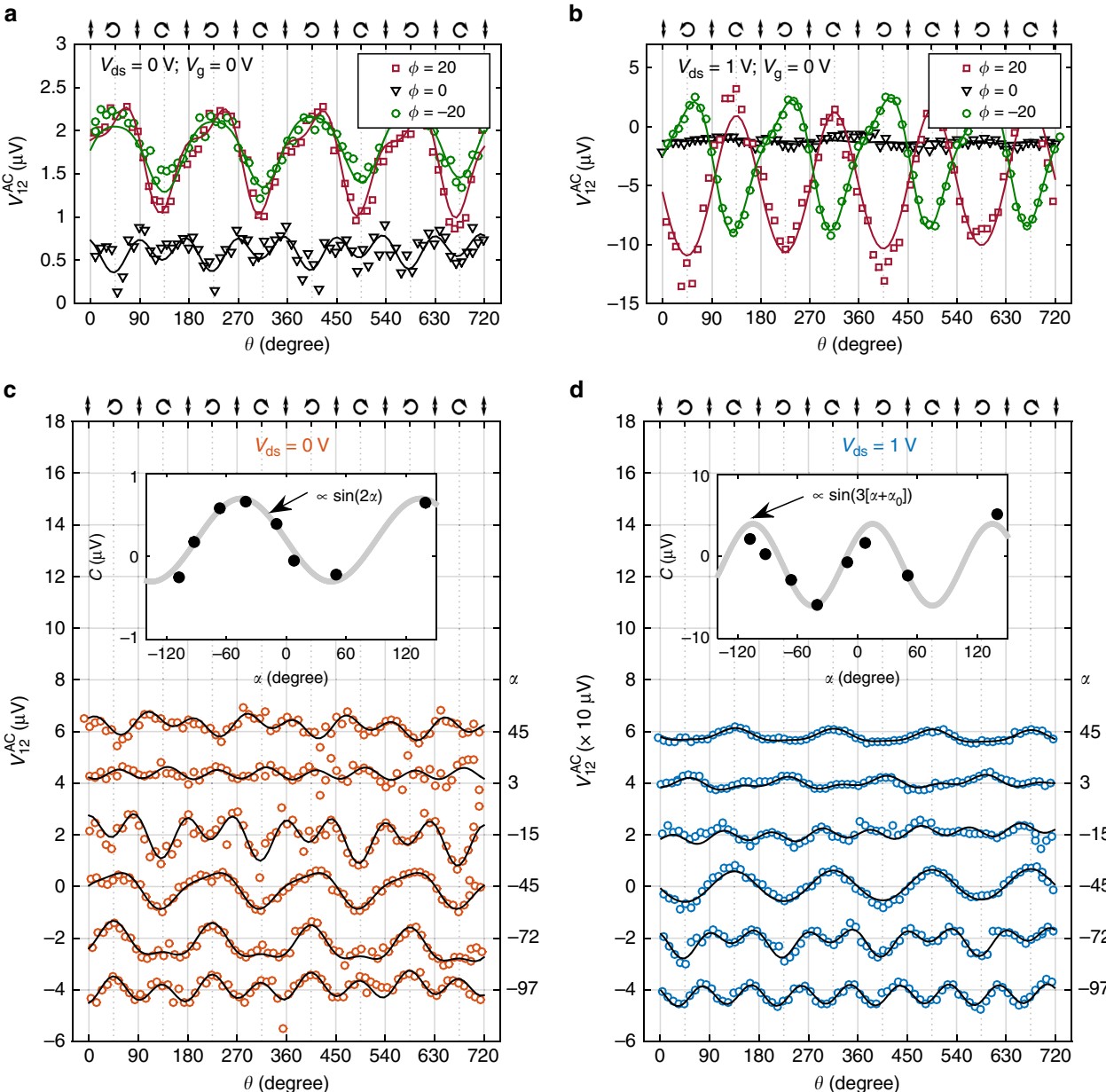

**Fig. 4** Effect of the illumination angle on the CPC amplitude, $C$. **a** Helicity-dependent photovoltage measured at $V_{ds} = 0$ V, $V_{gate} = 0$ V and $\lambda = 785$ nm for three different illumination incidence angles, $\phi = -20°$, $0°$ and $20°$. The azimuthal angle, $\alpha$, is fixed at $\alpha = 45°$. **b** Same as **a** for $V_{ds} = 1$ V. **c** Helicity-dependent photovoltage at different azimuthal angles, $\alpha$, with $V_{ds} = 0$ V and $\phi = 20°$. For clarity, the helicity-independent background has been removed and the plots have been shifted vertically in steps of 2 μV. Inset: CPC amplitude, $C$, extracted by fitting the measured photovoltage to Equation 1, as a function of $\alpha$. We observe that $C$ changes proportionally to $\sin(2\alpha)$. **d** Same as **c** for $V_{ds} = 1$ V. In this case, $C$ changes proportionally to $\sin(3\alpha)$

We further check the symmetry of the measured CPC by characterizing its dependence on the azimuthal angle $\alpha$ (see Fig. 1a). Figure 4c, d show the measured helicity-dependent photovoltages at different azimuthal angles, for $|V_{ds}| < V_T$ (Fig. 4c) and $|V_{ds}| > V_T$ (Fig. 4d). The insets show the dependence of $C$ on $\alpha$. Again, two different behaviors emerge: for small $V_{ds}$, $C$ is proportional to $\sin(2\alpha)$. We remark that, since the CPC sign is preserved upon inversion of $\phi$, it must also be preserved upon a $\pi$ rotation of $\alpha$ (both operations are equivalent in our system), and therefore, only a $\pi$-periodic dependence on $\alpha$ can appear.

For large $V_{ds}$, $C$ shows a modulation proportional to $\sin(3[\alpha + \alpha_0])$, where $\alpha_0$ is an angle offset (15° in our case). This $3\alpha$-periodic signal suggests that $C$ is modulated by the 120°-periodic crystal structure of 1L-MoSe$_2$. The presence of an angular offset $\alpha_0$ is

expected since the orientation of the crystal is not necessarily aligned with the electrodes. Oppositely from before, only an $\alpha$-dependence that gives an exact inversion upon $\pi$ increase of $\alpha$ can emerge, for consistency with signal inversion when $\phi$ is reversed.

As discussed in Supplementary Note 6, when the device symmetry is reduced to, at most, a single-mirror symmetry (which can be expected in a realistic device due to interface effects at the electrodes and in-plane strain gradients), a CPDE photocurrent can have a term proportional to $\sin(2\alpha)\sin^2(\phi)$, consistent with the observed behavior at low $V_{ds}$. For the large $V_{ds}$ regime, the inversion of the CPC upon sign flip of $\phi$ is consistent with both CPGE and CPDE terms (or a combination of them) allowed for this symmetry. Further, $\phi$-odd terms are also allowed for CPGE and CPDE under the more restrictive $C_{3v}$ symmetry.

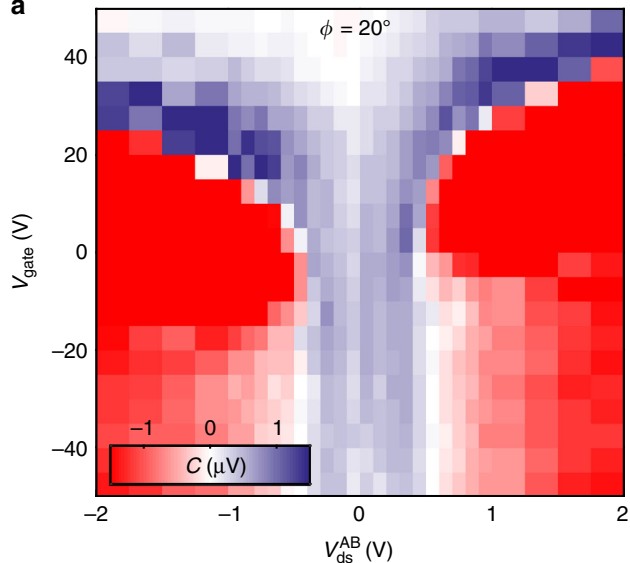

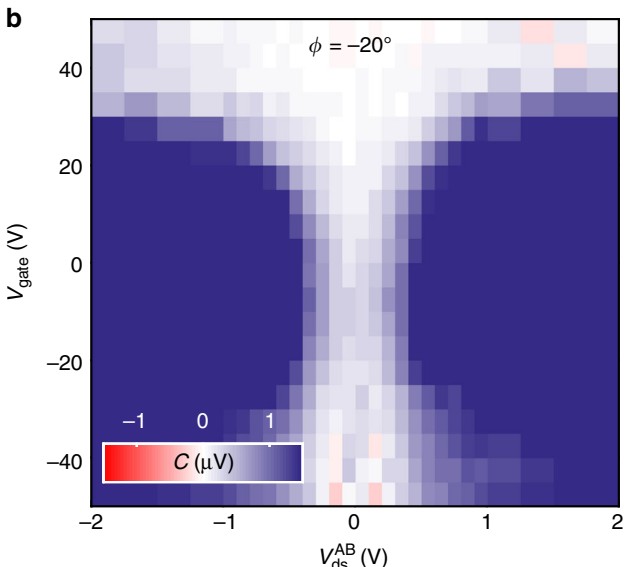

**Fig. 5** Effect of the gate voltage in the CPC. **a** Colormap of the CPC amplitude, $C$ for $\lambda = 785$ nm as a function of the drain-source and gate voltages, $V_{ds}$ and $V_{gate}$ for $\phi = 20°$. **b** Same as **a** for an incidence angle $\phi = -20°$

Notably, the dependence as $\sin(3\alpha)$ for the CPC measured at large $V_{ds}$ does not appear in the symmetry analysis. Such dependence, however, can emerge from inhomogeneities of the transport properties between the armchair and zigzag directions of the 1L-MoSe$_2$ crystal lattice, not considered in the theory. The symmetry analysis from Supplementary Note 6 also shows that a BC-CPGE can only appear for, at most, single-mirror symmetry, and even in that case, it should be maximal for illumination normal to the crystal plane, contrary to the observed absence of CPC in these conditions. Thus, we conclude that this effect does not yield a detectable contribution to our measurements.

**Effect of the gate voltage in the CPC.** Finally, we explore how the CPC is affected by the gate voltage. Figure 5a shows a color map of the CPC amplitude $C$ (derived from $V_{12}^{AC}$ lock-in signal) as a function of $V_{ds}$ (applied between electrodes $A$ and $B$) and $V_{gate}$, at $\alpha = -45°$ and $\phi = 20°$. The two drain-source voltage regimes

discussed above can be observed here as the blue ($C > 0$ mV) and red ($C < 0$ mV) areas of the map. Once again, when the incidence angle $\phi$ is changed to $-20°$ (see Fig. 5b), the sign of $C$ at large $V_{ds}$ switches from negative to positive, while at small $V_{ds}$ the sign is preserved. For $\phi = 0°$ (shown in Supplementary Note 4), we find that $C$ remains nearly zero regardless of the applied $V_{ds}$ and $V_{gate}$.

For $V_{gate}$ below 0 V we see a much weaker influence on the CPC than for $V_{gate} > 0$ V. In the latter case, we observe a shift of the transition voltage $V_T$ towards larger drain-source voltages. This can be explained by an increased trion population, due a higher density of charge carriers in the MoSe$_2$ crystal when the Fermi energy is brought above the edge of the conduction band[22]. Further, an increased gate voltage can also modify the electric field screening, changing the exciton and trion momentum lifetimes and therefore changing their contributions to the CPC[11,14]. When the gate voltage is further increased, we observe an overall reduction of the CPC, regardless of the value of $V_{ds}$, which we associate to a decrease of the carrier momentum lifetime, due to an enhanced electron–electron scattering. Also, the probability of exciton absorption is expected to decrease at large gate voltages, due to the reduced density of unoccupied states in the conduction band.

## Discussion
In conclusion, the two observed regimes for the CPC can be well described by CPGE and CPDE for a reduced device symmetry. Although effects of higher order in the light electric field could also be allowed by symmetry, the linearity of $C$ with illumination power confirms that the measured signal is dominated by second-order effects.

Importantly, although a BC-CPGE could be allowed for a low-symmetry device, it is not observed here, as confirmed by the fading of $C$ for incidence normal to the crystal plane. Further, our results indicate a transition from exciton-dominated to trion-dominated transport between the two regimes, but the influence of the excitonic character on CPC is an open question.

## Methods
**Device fabrication.** We mechanically exfoliate atomically thin layers of MoSe$_2$ and h-BN from their bulk crystals on a SiO$_2$ (300 nm)/doped Si substrate. The monolayer MoSe$_2$ and bilayer h-BN are identified by their optical contrasts with respect to the substrate[26] and their thickness is confirmed by atomic force microscopy (see Supplementary Note 1). Using a polymer-based dry pick-up technique, described in detail in ref.[27], we pick up the bilayer h-BN flake using a PC (poly(bisphenol A)carbonate) layer attached to a polydimethylsiloxane (PDMS) stamp. Then we use the same stamp to pick up the MoSe$_2$ flake directly in contact with the h-BN surface and we transfer the whole stack onto a bulk h-BN crystal, exfoliated on a different SiO$_2$/Si substrate. After the final transfer step, the PC layer is detached from the PDMS, remaining on top of the 2L-BN/MoSe$_2$/bulk-BN stack, and must be dissolved using chloroform. Next, to further clean the stack, we anneal the sample in Ar/H$_2$ at 350 °C for 3 h. For the fabrication of electrodes, we pattern them by electron-beam lithography using PMMA (poly(methyl methacrylate)) as the e-beam resist, followed by e-beam evaporation of Ti(5 nm)/Au(75 nm) at $10^{-6}$ mbar and lift-off in acetone at 40 °C.

**Electrical characterization.** The DC electrical characterization of the studied device is discussed in detail in Supplementary Note 3. The highly doped Si substrate is used as the back-gate electrode in order to tune the density of charge carriers in the MoSe$_2$ channel. To eliminate the effect of environmental adsorbates, all the electrical measurements are performed in vacuum (~$10^{-4}$ mbar). We measure the source-drain current as a function of the source-drain and back-gate voltages in four-terminal geometry of electrodes, using the side contacts of the Hall bars[28] as voltage probes. These measurements allow us to obtain a reliable estimation of conductivity and field effect mobility of charge carriers in the monolayer MoSe$_2$ channel. Further $I$–$V$ characteristics are measured in three-terminal geometry to evaluate quality of the electrical contacts at the metal–semiconductor interface, as further discussed in Supplementary Note 3.

**Data availability.** The data that support the findings of this study are available from the corresponding author upon request.

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

## Acknowledgements

We thank Feitze A. van Zwol, Tom Bosma, and Jakko de Jong for contributions to the laser control system. We also thank H. M. de Roosz, J. G. Holstein, H. Adema, and T. J. Schouten for technical assistance. This research has received funding from the Dutch Foundation for Fundamental Research on Matter (FOM), as part of the Netherlands Organization for Scientific Research (NWO), FLAG-ERA (15FLAG01-2), and the Zernike Institute for Advanced Materials.

## Author contributions

B.J.v.W. and C.H.v.d.W. initiated the project. J.Q. and T.S.G. had the lead in experimental work. J.-S.Y. and J.v.d.B. provided the theoretical analysis.

## Additional information

**Competing interests:** The authors declare no competing interests.

