## [Peer Review File · Nature Communications]

Reviewers' comments:

Reviewer #1 (Remarks to the Author):

The authors have reported the systematic investigation of helicity-dependent charge and spin photocurrents in a BN-embedded 1L MoSe₂ device. The circular photocurrents have been found to be dependent on diverse experimental factors such as device symmetry, excitation wavelength, incident angle, bias and gate voltages. The corresponding theoretical analysis was performed to identify the contributions from different physical origins such as circular photogalvanic and photon drag effects. Overall, the present work brings the timely knowledge of symmetry effects on photocurrents in 1L MoSe₂, which is quite interesting for the rising 2D semiconductor community. Thus, I would like to reconsider its publication after addressing the following concerns.

1. For the data in Figure 1c, the authors should clearly describe that the amplitude was extracted from which two contacts and at which incident angle. Why the contributions from three components to the total photocurrents change with the excitation power?
2. In Figure 2b, the two main bands were marked as A₀ and B_{+/-}, respectively. The possible roles of A trions, dark A and B excitons should be discussed to be consistent with their previous publication (Ref. 19). Particularly, the dark excitons affected by the symmetry of excitonic states (e.g. Phys. Rev. Lett. 119, 047401, 2017) may contribute to the observed photocurrents at non-normal incidence angles. Moreover, in Figure 3c, they claimed that the dominant charge carriers at low drain-source voltages (e.g. $V_{ds} = 0$) were A trions. Thus, it is hard to claim that the main photocurrent peak and the CPC contribution C in Figure 2b are simply resonant with A₀ exciton rather than A trion.
3. The electrical characterization of the device indicated that the used 1L MoSe₂ is a n-type material, where the positive trions hardly exist in general. The claimed roles of A⁺ and B⁺ in the observed photocurrents at $V_{gate} = 0V$ may need to be reconsidered.
4. During the session on "Effect of the gate voltage in the CPC", C values extracted at a fixed excitation wavelength of 785 nm were plot in Figure 5. It will be better to further clarify such effect if the authors can provide the CPC amplitude of C as a function of the illumination wavelength (like Figure 3c) at varied gate voltages (e.g. $V_g = 0$ and 40 V). If the C maxima change with the gate voltages rather than remaining at a fixed wavelength (e.g. 785 nm), the corresponding discussion should be refined.
5. The novelty of the present work should be further highlighted to be beyond previous studies (e.g. Refs. 10, 11 and 19).

Reviewer #2 (Remarks to the Author):

In this work, the authors study experimentally photocurrents in monolayer MoSe₂ which are then analyzed under a symmetry perspective to isolate the different angle dependence that can contribute to the measured photovoltages. They conclude that there are two different regimes as a function of source drain voltage with different symmetry properties. At low voltages, the photovoltages are consistent with a single mirror plane at most while at high voltages, C_{3v} or less symmetry at high voltages. The authors interpret the two regimes to be related to different types of excitons, although the connection between these two is not firmly established and left as an open question. They conclude as well that the circular photogalvanic Berry curvature term is not responsible of any of this effects.

I find this work interesting and timely. Non-linear effects in general, and second order effects in particular are experiencing a revival both in single and few layer materials as well as topological metals. It is also clearly written, with an adequate style appropriate to the journal.

However I would like the authors to clarify a few aspects of their interpretation.

My main concern is the Berry curvature contribution. In particular, the Fermi golden rule can be

reduced to the Berry curvature only in the two band approximation, as far as I understand from Ref.10 of the supplementary material. Else there are higher band corrections scale as $\omega^2/\Delta E^2$ where ΔE is the distance to other bands. TMDs have known descriptions in terms of four or three band models so I would expect these corrections not to be negligible, unless matrix elements render them zero. Although these considerations don't change the conclusions on the angular dependence I think they do affect the generality of the claim that the Berry curvature does not contribute. In particular since the $\cos(\phi)$ dependence would not only come from Berry curvature strictly, but from Berry curvature + corrections one would have to assess the size of these corrections to make the claim, which I believe is not addressed in the current version.

Also, can the authors discard intraband effects? It is not entirely clear to me if the response is purely interband; finite frequency interband effects can peak at frequencies of the order of the gate voltage doping. The authors do not discuss intraband contributions, which in non-linear effects can depend strongly in frequency (see 1803.02850v1). Possibly these have the same angular dependence, so can affect the conclusion mentioned in my first point.

Finally, the authors mention the different excitonic contributions as a possibly relevant explanation of the two regimes, which, as far as I could tell is based on Figure 3c and reference 22. I think this is quite an interesting connection, which I would like the authors to discuss further. For instance, a more self contained explanation of the type of excitons mentioned in the main text can be useful for the inexperienced reader as a supplementary information section.

Reviewer #3 (Remarks to the Author):

This paper reports measurements of photocurrent and photovoltage under different conditions of photoexcitation. The main point is that the Berry curvature contribution is not seen, as evidenced by the non-observation of a CPGE effect when the light is at normal incidence. This conclusion is in contrast with a previous report in Nat. Comm. [Ref. 11 of the main text] in which the current was associated with the Berry-related anomalous velocity. Ref. 12 is also cited as reporting CPGE in 1L-TMDC's, although this does not seem to be the subject of that paper.

Overall, although many interesting effects are seen, few are actually explained. The various dependences on source-drain and gate voltage, as well as the dependence on the angle of the plane of incidence all fall into this category. It appears that the only sharp statement offered in the paper is the one mentioned above, namely that the intrinsic Berry-related effect is not seen. While this is certainly a valid and meaningful contribution, it is hard to assess its overall impact.

What I believe is missing from the present version of the paper is a better picture of the magnitude of the photocurrent. In the abstract, the authors make reference to devices. Are there aspects of the current in this system that make it particularly attractive for certain specific applications? Another point related to the magnitude is the following. If I understand the paper correctly, the Berry related CPGE at normal incidence is allowed by symmetry. Thus, based on general principles, it should be there. Furthermore, given its intrinsic nature, it should be possible to estimate its magnitude. Again, if I understand the supplement, the only extrinsic parameter involved is the carrier lifetime, τ . What I would be looking for then, based on estimate of the intrinsic current, is whether it would be detectable in the present experiment. If Berry-related CPGE exists in this system, but is small relative to other contributions, then presumably it would induce a small shift in the angle at which zero current is observed, and this small shift might be difficult to detect.

Overall, I believe the paper would be strengthened if more quantitative statements could be made concerning the absolute magnitudes of the various effects and their origin. Finally, is it possible that most or all of the observed currents are not strictly intrinsic but instead are related to the

asymmetric scattering from impurities that becomes allowed in non-centrosymmetric environments.

Reviewer #1:

The authors have reported the systematic investigation of helicity-dependent charge and spin photocurrents in a BN-embedded 1L-MoSe₂ device. The circular photocurrents have been found to be dependent on diverse experimental factors such as device symmetry, excitation wavelength, incident angle, bias and gate voltages. The corresponding theoretical analysis was performed to identify the contributions from different physical origins such as circular photogalvanic and photon drag effects. Overall, the present work brings the timely knowledge of symmetry effects on photocurrents in 1L MoSe₂, which is quite interesting for the rising 2D semiconductor community. Thus, I would like to reconsider its publication after addressing the following concerns.

1.1. For the data in Figure 1c, the authors should clearly describe that the amplitude was extracted from which two contacts and at which incident angle.

In the revised version of the manuscript, Figure 1c explicitly indicates the set of electrodes for which the amplitude was measured (contacts 1 and 2). For further clarification, the supporting information now also includes a plot (see below) of the power dependence of C, L₁ and L₂ for a different set of electrodes (A and B) and measuring photocurrent instead of photovoltage:

Figure S7 shows the power dependence of C, L₁ and L₂ for electrodes A and B and measuring photocurrent instead of photovoltage, in contrast with the measurements shown in the main text for contacts 1 and 2 (Figure 1c). This shows that the same linear power dependence appears regardless of the contacts or the measurement technique.

Fig S7 Illumination power dependence of L₁, L₂ and C extracted from the fittings of helicity-dependent photocurrent measurements (as those shown in Fig S6) for contacts A and B.

Why the contributions from three components to the total photocurrents change with the excitation power?

For second order effects, the signal must be proportional to E^2 (or, equivalently, to the excitation power), where E is the amplitude of the light electric field. Thus, the fact that C , L_1 and L_2 change linearly with the excitation power indicates that they correspond to a second order response to the light electric field (in particular, photogalvanic and photon drag effects). This is now clarified in the main text:

“The three amplitudes increase linearly with the illumination power P , indicating that they are due to a second order response to the light electric field E (and thus, proportional to $E^2 \propto P$), in agreement with earlier literature for 1L-MoS₂.^[1]”

1.2. In Figure 2b, the two main bands were marked as A0 and B+/-, respectively. The possible roles of A trions, dark A and B excitons should be discussed to be consistent with their previous publication (Ref. 19). Particularly, the dark excitons affected by the symmetry of excitonic states (e.g. Phys. Rev. Lett. 119, 047401, 2017) may contribute to the observed photocurrents at non-normal incidence angles.

The revised manuscript includes additional discussion addressing this point:

“We also remark that, as shown in recent experiments, dark exciton transitions in TMDCs can become optically active under oblique illumination²³. Further, earlier work by our team showed that for a non-encapsulated 1L MoSe₂ device dark-exciton contributions can even appear under illumination perpendicular to the crystal plane¹⁹, giving an additional peak in the PC spectrum at an energy ~ 30 meV above the A⁰ resonance, i.e. at ~ 775 nm. This feature is not observed with the h-BN encapsulated device studied here, suggesting that the asymmetric interaction with SiO₂ could be relevant to opening the dark exciton optical transition. h-BN substrates are known to yield a strengthened A⁰ exciton absorption^[2], which could mask a weak dark exciton contribution in the present work.”

Moreover, in Figure 3c, they claimed that the dominant charge carriers at low drain-source voltages (e.g. $V_{ds} = 0$) were A trions. Thus, it is hard to claim that the main photocurrent peak and the CPC contribution C in Figure 2b are simply resonant with A0 exciton rather than A trion.

We thank the reviewer for raising this valid point. This was indeed an imprecision in the text and has been corrected in the revised version (page 6):

“Figure 2b shows the wavelength dependence of C , L_1 and L_2 . The CPC contribution C is maximal when the illumination is on-resonance with the A exciton or trion transitions ($\lambda = 785\text{--}795$ nm) and progressively decreases when the illumination is shifted away from the resonance”

1.3. The electrical characterization of the device indicated that the used 1L MoSe₂ is an n-type material, where the positive trions hardly exist in general. The claimed roles of A⁺ and B⁺ in the observed photocurrents at $V_{\text{gate}} = 0$ V may need to be reconsidered.

We agree with the reviewer. In the original text, we intended to keep open the possibility of A⁺ and B⁺ trion contributions, however, for an n-doped material, A⁻ and B⁻ trions are expected to dominate. This is specifically clarified in the revised manuscript:

“Note that, given the n-type behavior of the 1L-MoSe₂ channel, negatively charged trions, A⁻ are expected to have a much larger contribution to the CPC than their positive counterparts A⁺.”

1.4. During the section on “Effect of the gate voltage in the CPC”, C values extracted at a fixed excitation wavelength of 785 nm were plot in Figure 5. It will be better to further clarify such effect if the authors can provide the CPC amplitude of C as a function of the illumination wavelength (like Figure 3c) at varied gate voltages (e.g. $V_g = 0$ and 40 V). If the C maxima change with the gate voltages rather than remaining at a fixed wavelength (e.g. 785 nm), the corresponding discussion should be refined.

Following the advice of the referee, we have performed additional measurements and spectral characterization on the same sample for different combinations of V_{ds} and V_g . The results are discussed in the revised Supporting Information (section S7):

“Figure S8 shows the CPC amplitude C as a function of the illumination wavelength for different combinations of voltages, V_{ds} and V_g . The resonant character of C is clearly observed for all measurements, with the maximum signal occurring at 785 nm for $V_{ds} = 1$ V and at 790 nm for $V_{ds} = 0$ V (see also Fig. 3c in the main text).

For the data acquired at $V_{ds} = 1$ V, a weaker but nonzero CPC is also observed for off-resonance excitation with energies above the 1L-MoSe₂ absorption edge, which could be associated to the emergence of free-electron driven CPC. At negative gate voltages, this CPC contribution even presents a different sign from that of the main peak at ~785 nm. Although relevant, a comprehensive analysis of this off-resonance CPC contribution is beyond the scope of our present work.”

Figure S8 – Spectral dependence of C at $V_g = 40\text{ V}$, -20 V and -40 V , as labelled, for $V_{ds} = 1\text{ V}$ (a) and $V_{ds} = 0\text{ V}$ (b). The spectral measurements corresponding to $V_g = 0$ can be found in the main text (Figure 3c). The dashed vertical lines are guides to the eye for 785 nm (green) and 790 nm (orange).

The discussion of this point in the main text was also slightly modified for consistency with the new data

“Figure 3c shows the CPC amplitude C as a function of the wavelength at $V_{ds} = 0\text{ V}$ and $V_{ds} = 1\text{ V}$. Interestingly, the wavelength at which C is maximized (in absolute value) for $V_{ds} = 0\text{ V}$ occurs at a wavelength 5-10 nm longer (in energy 10-20 meV lower) than that obtained for $V_{ds} = 1\text{ V}$. A similar spectral shift is also observed for different gate voltages (see Suppl. Info. S7). The observed shift is consistent with recent photoluminescence experiments that showed a gate-induced exciton-to-trion transition in a monolayer TMDC.^[3] This suggests that at low drain-source voltages the dominant charge carriers involved in the CPC are $A^{\pm/}$ trions (which have a nonzero charge and therefore do not need to dissociate to participate in the photovoltage), while at large drain-source voltages the transport is dominated by dissociated A^0 excitons. A similar spectral response is also observed for different gate voltages (see Suppl. Info. S7)”

1.5. The novelty of the present work should be further highlighted to be beyond previous studies (e.g. Refs. 10, 11 and 19).

Following the advice of the reviewer we modified the introduction paragraph to further stress the novelty of our work as compared with recent literature. The relation with reference 19 is addressed elsewhere in the text (see question 1.2 above). The new introduction reads as follows (underlined parts are modified from the previous version):

“In this work we investigate the spectral and electrical behavior of the helicity-dependent circular photocurrent (CPC) in a 1L-TMDC, providing a comprehensive experimental characterization of this effect. In an earlier work^[1], it was suggested that exciton transitions could play a role in the generation of CPC. Here, by evaluating the spectral response of the CPC in a h-BN encapsulated 1L-MoSe₂ phototransistor, we show that the CPC amplitude is maximized when the illumination wavelength matches the A exciton resonance, clearly confirming the excitonic character of CPC in 1L-TMDCs. In another recent experiment on multilayer WSe₂^[4] it was shown that the strength of the CPC response could be changed by a gate voltage, but the effect of the drain-source voltage V_{ds} was not studied. Our study here includes the dependence of the CPC on V_{ds} , revealing two different regimes for small (below 0.4 V) and large voltages, with the CPC changing sign between one regime and the other. At certain fixed V_{ds} this CPC sign change can also be induced via V_g . Further, by testing the dependence of CPC on the light incident angle we find that it presents very different symmetry for the two regimes: For small V_{ds} , the CPC is preserved when the incidence angle is switched from ϕ to $-\phi$, whereas for large V_{ds} , inverting the illumination angle ϕ causes a change of sign for the CPC, pointing to a separate physical origin. In ref.^[1] it was proposed that Berry-curvature (BC) at the band edges of 1L-TMDCs can give a contribution to CPC (BC-induced circular photogalvanic effect, BC-CPGE). However, the expected dependence of this effect on the light incidence angle is not compatible with the angular dependences observed here for any of the two CPC regimes. Thus, we conclude that BC-CPGE can be ruled out as a dominant mechanism involved. Further, we show that CPC can also emerge in this system due to the circular photon drag effect (CPDE), mostly overlooked in prior literature for 1L-TMDCs. Finally, we show how by applying a gate voltage to modify the Fermi energy of the 1L-MoSe₂ channel, one can tune the relative strength of the two contributions at a fixed drain-source voltage, achieving control over the intensity and direction of the helicity-dependent photoresponse.”

Reviewer #2:

In this work, the authors study experimentally photocurrents in monolayer MoSe₂ which are then analyzed under a symmetry perspective to isolate the different angle dependence that can contribute to the measured photovoltages. They conclude that there are two different regimes as a function of source drain voltage with different symmetry properties. At low voltages, the photovoltages are consistent with a single mirror plane at most while at high voltages, C_{3v} or less symmetry at high voltages. The authors interpret the two regimes to be related to different types of excitons, although the connection between these two is not firmly established and left as an open question. They conclude as well that the circular photogalvanic Berry curvature term is not responsible of any of this effects.

I find this work interesting and timely. Non-linear effects in general, and second order effects in particular are experiencing a revival both in single and few layer materials as well as topological metals. It is also clearly written, with an adequate style appropriate to the journal.

However I would like the authors to clarify a few aspects of their interpretation.

2.1. My main concern is the Berry curvature contribution. In particular, the Fermi golden rule can be reduced to the Berry curvature only in the two band approximation, as far as I understand from Ref.10 of the supplementary material. Else there are higher band corrections scale as $\omega^2/\Delta E^2$ where ΔE is the distance to other bands. TMDs have known descriptions in terms of four or three band models so I would expect these corrections not to be negligible, unless matrix elements render them zero. Although these considerations don't change the conclusions on the angular dependence I think they do affect the generality of the claim that the Berry curvature does not contribute. In particular since the $\cos(\phi)$ dependence would not only come from Berry curvature strictly, but from Berry curvature + corrections one would have to assess the size of these corrections to make the claim, which I believe is not addressed in the current version.

We thank the reviewer for raising this point. Indeed, higher band corrections do not change the conclusions on the angular dependence by symmetry analysis. Ref. 10 discussed the higher band corrections for more general multi-band models. In general, the CPGE is a complicated process when the photon energy is large. The corrections could be computed by first principles. The revised Suppl. Info. includes a discussion about corrections arising from additional bands to make this point more explicitly:

"For multi-band cases, $\Omega_I^z(\vec{k}) \neq \Omega_F^z(\vec{k})$ and we could have additional corrections from other bands to the Berry curvature [4] of the form

$$-2 \sum_{n \neq F, I} \frac{\text{Im} \left(\langle u_{n\vec{k}} | \frac{\partial \hat{H}}{\partial k_x} | u_{I\vec{k}} \rangle \langle u_{I\vec{k}} | \frac{\partial \hat{H}}{\partial k_y} | u_{n\vec{k}} \rangle \right)}{[E_n(\vec{k}) - E_I(\vec{k})]^2} \quad (1)$$

In pristine monolayer TMDCs, the conduction and valence band edges can be well-described by a two-band (without the inclusion of spins) or four-band (with the inclusion of spins), massive Dirac fermion model (see Phys. Rev. Lett. 108, 196802 (2012) and for a recent review, Nat. Phys. 10, 343 (2014)). In this effective model (supported by first principle calculations), the Berry curvatures at the conduction and valence bands take opposite values, $\Omega_F^z(\vec{k}) = -\Omega_I^z(\vec{k})$, where F stands for the conduction bands (CB) and I for the valence band (VB). The Berry curvature is dependent on spin through the spin-dependent band gap. In this description, the Fermi golden rule with two-band approximation allows us to connect the photocurrent to the Berry curvature with no additional corrections. TMDCs with lower symmetry are well described by a tilted Dirac model (see Phys. Rev. Lett. 115, 216806 (2015)) and the Berry curvature is also found to take opposite values for conduction and valence band edges in this case. However, first principle calculations are needed for understanding the higher band corrections to the photocurrent.”

Note that, for the excitation wavelengths considered in this work, optical transitions only occur near the conduction and valence band edges, which can be well described by a massive Dirac fermion model (Phys. Rev. Lett. 108, 196802 (2012) and Nat. Phys. 10, 343 (2014).

2.2. Also, can the authors discard intraband effects? It is not entirely clear to me if the response is purely interband; finite frequency interband effects can peak at frequencies of the order of the gate voltage doping. The authors do not discuss intraband contributions, which in non-linear effects can depend strongly in frequency (see 1803.02850v1). Possibly these have the same angular dependence, so can affect the conclusion mentioned in my first point.

We acknowledge that the reviewer raises here a valid question. The intraband effects discussed in 1803.02850v1 are indirect effects, which involve both a direct optical transition from conduction to valence band and a process of impurity scattering from valence to conduction band.

This intraband contributions can only occur for $\hbar\omega < 2\mu$, where $\hbar\omega$ is the the photon energy and μ is the chemical potential (choosing $\mu = 0$ at the bottom of the conduction band). In our study, the photon energy is around 1.6 eV, which is much larger than the chemical potential, in the order of 0.1-0.2 eV even when a gate voltage is applied. This indirect intra-band effect could be observed in infrared to microwave regime, but does not play a role in our measurements.

This point is now addressed on page 17 of the Supplementary Information:

“One could also consider a possible role for intraband transitions, as discussed in ref. ^[5]. However, these intraband contributions can only occur for $\hbar\omega < 2\mu$, where $\hbar\omega$ is the the photon energy and μ is the chemical potential (choosing $\mu = 0$ at the bottom of the conduction band). In our measurements, the photon energy is around 1.6 eV, well above the chemical potential, in the order of 0.1-0.2 eV even when

a gate voltage is applied. Thus, such effects are not expected to give a strong contribution here, but could still be observed in infrared to microwave regime.”

2.3. Finally, the authors mention the different excitonic contributions as a possibly relevant explanation of the two regimes, which, as far as I could tell is based on Figure 3c and reference 22. I think this is quite an interesting connection, which I would like the authors to discuss further. For instance, a more self-contained explanation of the type of excitons mentioned in the main text can be useful for the inexperienced reader as a supplementary information section.

In response to this point, as well as similar comments raised by Reviewer 1, we have extended and improved the discussion on the connection between CPC and exciton transitions (see questions 1.2, 1.3 and 1.4). Further, as a help for the inexperienced reader, the revised Supplementary Information includes a brief section commenting on the different valley exciton transitions for TMDCs:

“S8. Brief notes on valley exciton transitions in monolayer MoSe₂

The diagram shown in Figure S9 summarizes the different exciton absorption mechanisms that can occur for monolayer MoSe₂. For each pair of spin-orbit split subbands one gets an optically active neutral exciton (A⁰ and B⁰) which, at room temperature, are expected to give absorption peaks at 1.58 eV and 1.78 eV. Positively and negatively charged trion absorption can also occur (A^{+/-} and B^{+/-}). Finally, electrons and holes from subbands with opposite spin can also combine to form the so-called dark excitons (A_D⁰ and B_D⁰). Dark exciton absorption is a priori spin-forbidden, but can become allowed for oblique illumination or even for normal incidence in the presence of a gate voltage. Further review on exciton physics in TMDCs can be found in references ^[6] and ^[7]. “

Figure S9 - Diagram illustrating the possible valley exciton transitions for monolayer MoSe₂.

Reviewer #3:

In order to respond to Reviewer 3 in a structured way, we arranged their comments in numbered sections and answered each one separately.

3.1. This paper reports measurements of photocurrent and photovoltage under different conditions of photoexcitation. The main point is that the Berry curvature contribution is not seen, as evidenced by the non-observation of a CPGE effect when the light is at normal incidence. This conclusion is in contrast with a previous report in Nat. Comm. [Ref. 11 of the main text] in which the current was associated with the Berry-related anomalous velocity. Ref. 12 is also cited as reporting CPGE in 1L-TMDC's, although this does not seem to be the subject of that paper.

Ref. 12 was cited here because it contains the theoretical background in which the claims of ref. 11 are supported. For clarity, we removed the reference from that paragraph, as it is also cited elsewhere in the text.

3.2. Overall, although many interesting effects are seen, few are actually explained. The various dependences on source-drain and gate voltage, as well as the dependence on the angle of the plane of incidence all fall into this category. It appears that the only sharp statement offered in the paper is the one mentioned above, namely that the intrinsic Berry-related effect is not seen. While this is certainly a valid and meaningful contribution, it is hard to assess its overall impact.

Indeed, the microscopic mechanisms giving rise to the observed CPC are yet unclear (both here and in related recent literature), and require further investigation. However, this work provides the most complete and consistent phenomenological description of this effect to date, including the spectral characterization of CPC in TMDCs (never shown before in literature and clearly confirming the excitonic character of CPC), a comprehensive electrical characterization of CPC in TMDCs (which shows for the first time the existence of at least two separate voltage regimes for CPC), and a complete characterization of the angular dependence of CPC. This latter aspect was never performed before in such depth, and we include the first theoretical analysis of CPC in 2D TMDCs by symmetry arguments. Our results allow to reduce the range of possible mechanisms responsible for CPC and rule out the previous understanding of this effect as mainly caused by Berry phase induced CPGE. Thus, we firmly believe that our work will have a high impact in the field, and will surely be a valuable resource for further investigation of CPC in 2D-TMDCs.

3.3. What I believe is missing from the present version of the paper is a better picture of the magnitude of the photocurrent

We elaborate on this in our answer to point 5.

3.4. In the abstract, the authors make reference to devices. Are there aspects of the current in this system that make it particularly attractive for certain specific applications?

We introduced a small modification to the introduction to make this point more explicit (page 2):

“(…) the CPGE allows to generate a directed spin-valley current even without applying any voltage, bringing new opportunities for the implementation of spintronic and valleytronic devices where the direction and intensity of spin and valley currents can be controlled using light only.”

3.5. Another point related to the magnitude is the following. If I understand the paper correctly, the Berry related CPGE at normal incidence is allowed by symmetry. Thus, based on general principles, it should be there. Furthermore, given its intrinsic nature, it should be possible to estimate its magnitude. Again, if I understand the supplement, the only extrinsic parameter involved is the carrier lifetime, τ . What I would be looking for then, based on estimate of the intrinsic current, is whether it would be detectable in the present experiment. If Berry-related CPGE exists in this system, but is small relative to other contributions, then presumably it would induce a small shift in the angle at which zero current is observed, and this small shift might be difficult to detect. Overall, I believe the paper would be strengthened if more quantitative statements could be made concerning the absolute magnitudes of the various effects and their origin.

The original manuscript may have led to a misunderstanding here: According to our symmetry analysis, the BC-CPGE can only appear for MoSe₂ if the crystal symmetry is reduced by external agents – such as by mechanical strain, as well as thermal or electrical gradients. For the symmetry of the pristine MoSe₂ crystal, the discrete rotational symmetry of the system requires this effect to cancel out. Due to this dependence of BC-CPGE on external parameters like the magnitude of strain, providing a meaningful estimation of the order of magnitude is beyond our current level of sample control and characterization. This point is clarified in the revised manuscript:

“The symmetry analysis from Suppl. Info. section S6 also shows that a BC-CPGE can only appear for, at most, single-mirror symmetry. Even in that case, it should be maximal for illumination normal to the crystal plane, contrary to the observed absence of CPC in these conditions. Thus, we conclude that this effect does not yield a detectable contribution to our measurements.”

As discussed above, at the current state of the art, the origin of CPC in monolayer TMDs remains unclear, and a full description of this effect will require extensive further investigation, well beyond the scope of this work.

3.6. Finally, is it possible that most or all of the observed currents are not strictly intrinsic but instead are related to the asymmetric scattering from impurities that becomes allowed in non-centrosymmetric environments?

As suggested by the referee, asymmetric scattering could indeed produce polarization-dependent currents. In fact, as we explicitly state in the new version of the text, we believe that this is a possible explanation for the linear polarization-dependent signals L_1 and L_2 :

“For the linear photovoltage L a nonzero amplitude appears, also for out-of-resonance illumination. Further, we observed that the spectral dependence of L markedly changes between different sets of electrodes, even in the same 1L-MoSe₂ flake. The origin of a nonzero L is usually associated with scattering of the charge carriers with anisotropic local defects.²⁰ “

However, this behavior is not observed for the helicity-dependent signal C , which vanishes at normal incidence, and clearly has its strongest appearance for excitation energies in resonance with exciton transitions. Note that for the case of L , a nonzero contribution is observed even for non-resonant excitation (figure 2b) and normal incidence (figure 4a).

REVIEWERS' COMMENTS:

Reviewer #1 (Remarks to the Author):

The manuscript has been improved in view of the refined explanation and supporting data. All my comments have been taken into account and replied in detail. Meanwhile, the novelty has been further addressed. The present version indeed tells a better story. Overall, I recommend its publication considering their substantial revision and careful clarification.

Reviewer #2 (Remarks to the Author):

I thank the authors for their replies to my comments, which I find satisfactory. Examining all additions made to address the other referees comments as well, I recommend publication in Nat. Communications.

Reviewer #3 (Remarks to the Author):

I recommend acceptance of this paper in view of the response and revisions of the authors.

We thank all the reviewers for their useful and constructive feedback on our manuscript, as it led to an improvement of the overall quality and clarity of this work.

REVIEWERS' COMMENTS:

Reviewer #1 (Remarks to the Author):

The manuscript has been improved in view of the refined explanation and supporting data. All my comments have been taken into account and replied in detail. Meanwhile, the novelty has been further addressed. The present version indeed tells a better story. Overall, I recommend its publication considering their substantial revision and careful clarification.

Reviewer #2 (Remarks to the Author):

I thank the authors for their replies to my comments, which I find satisfactory. Examining all additions made to address the other referees' comments as well, I recommend publication in Nat. Communications.

Reviewer #3 (Remarks to the Author):

I recommend acceptance of this paper in view of the response and revisions of the authors.